# Cell Therapies in Bladder Cancer Management

**DOI:** 10.3390/ijms22062818

**Published:** 2021-03-10

**Authors:** Lucia Morales, Jesús M. Paramio

**Affiliations:** 1Molecular Oncology Unit, Centro de Investigaciones Energéticas, Medioambientales y Tecnológicas (CIEMAT), Ave Complutense 40, 28040 Madrid, Spain; 2Biomedical Research Institute I+12, University Hospital “12 de Octubre”, 28041 Madrid, Spain; 3Centro de Investigación Biomédica en Red de Cáncer (CIBERONC), 28029 Madrid, Spain

**Keywords:** bladder cancer, immunotherapy, chimeric antigen receptor, cell therapy, CAR-T cells

## Abstract

Currently, bladder cancer (BC) represents a challenging problem in the field of Oncology. The high incidence, prevalence, and progression of BC have led to the exploration of new avenues in its management, in particular in advanced metastatic stages. The recent inclusion of immune checkpoint blockade inhibitors as a therapeutic option for BC represents an unprecedented advance in BC management. However, although some patients show durable responses, the fraction of patients showing benefit is still limited. Notwithstanding, cell-based therapies, initially developed for the management of hematological cancers by infusing immune or trained immune cells or after the engineering of chimeric antigen receptor (CAR) expressing cells, are promising tools to control, or even cure, solid tumors. In this review, we summarize recent cell-based immunotherapy studies, with a special focus on BC.

## 1. Introduction

Bladder cancer (BC) is the fourth-leading type of cancer for estimated new cases in males in the U.S. in 2021 and eighth in estimated death cases [1]. The estimated total number of new BC cases in 2021 will be close to 84,000, with around 20% patient mortality. These dramatic numbers highlight the urgency of finding new and effective BC therapies.

BC therapies for advanced BC remained unchanged for several decades until the recent arrival of immune checkpoint inhibitors (ICI), which represent an unprecedented advance in the management of this type of cancer. Indeed, the use of immune reactivation is not new in BC, as in high-risk non-muscle-invasive bladder cancer (NMIBC) patients, the treatment includes intravesical instillation with *Bacillus Calmette–Guerin* (BCG) after transurethral resection. This treatment, which has become the gold standard for NMIBC since the 1970s, produces a local inflammatory response, mainly driven by the innate immune system, which prevents recurrences and progression of NMIBC. Although NMIBC shows a favorable prognosis, it also displays one of the highest incidences of recurrence (60–70%) and, in some cases, progression into muscle-invasive disease [2]. These NMIBC recurrence rates require thorough monitoring after treatment for an extended period which is associated with a high cost for health care systems. The options for these stages were less effective before the introduction of ICI-based therapies. Muscle invasive bladder cancer (MIBC) is usually treated by cisplatin-based neoadjuvant chemotherapy followed by radical cystectomy [3]. For highly selected patients, a less aggressive partial cystectomy followed by chemoradiation is an alternative that may provide similar oncologic outcomes while maintaining bladder and sexual functions [4]. European association of urology (EAU) guidelines on muscle-invasive and metastatic BC consider adjuvant chemotherapy treatment after surgery if patients have not received previous neoadjuvant treatment [5]. Recently, neoadjuvant dose-dense methotrexate, vinblastine, doxorubicin, and cisplatin (MVAC) treatment demonstrated improved survival rates in patients with locally advanced BC as compared with gemcitabine and cisplatin (NCT01031420) [6]. This treatment is highly aggressive and, due to other comorbidities associated with advanced age, in some cases, it cannot be used and only in few situations leads to complete pathological responses. Moreover, MIBC relapse and progression to metastatic disease occurs often and is associated with poor prognosis, and adjuvant chemotherapy only shows minor increases in patient survival [7]. All these clinical characteristics make perioperative immunotherapy an attractive option to be offered in clinical trial settings. In recent years the improvement of ICI-based immunotherapies in other solid tumors finally led to the approval of these therapeutic agents for BC management [8]. In platinum-relapsed patients with metastatic urothelial carcinoma, immunotherapy treatment using the ICIs pembrolizumab or atezolizumab are second-line treatment options [9,10], though while durable responses have been observed, the fraction of patients showing objective benefit is low, and there is ample room for increasing effectiveness. In particular, refractory metastatic urothelial carcinoma would greatly benefit from the development of new therapies. Novel treatment options for these patients were approved by Food and Drug Administration (FDA) and are currently under clinical investigation. Erdafitinib is a pan-fibroblast growth factor receptor inhibitor that targets this signaling pathway involved in BC tumorogenesis, and enfortumab vedotin is an antibody-drug conjugate (ADC) therapy that recognizes bladder cancer cells to deliver cytotoxic drugs [11].

In this review, we summarize immunotherapy studies carried out in BC. Since most of the immunotherapy treatments for BC patients are non-cell-based, we review those widely used therapies. However, taking into account that non-cell-based immunotherapies fail in some patients, cell-based immunotherapies are being developed as an alternative for the treatment of those BC patients. We discuss immunotherapy using innate and adaptive immune cells with a special focus on engineering chimeric antigen receptor (CAR)-T lymphocytes (T cells) and their improvement as a tool to cure BC.

## 2. Non-Cell-Based Immunotherapies

Deep knowledge of the immune system and its role in fighting cancer is essential for the development of cancer immunotherapies. Different non-cell-based immunotherapies have been tested, such as cytokines, immune-modulating drugs, vaccines, and antibodies (monoclonal or drug-conjugates) [12]. Here, we will discuss ICI treatment since it is being performed on BC patients, although it is not a cell-based immunotherapy.

ICI are monoclonal antibodies that block immune checkpoint proteins, which prevent the immune evasion of cancer cells [13]. In particular, cytotoxic T lymphocyte antigen 4 (CTLA-4) is an immune checkpoint molecule expressed in T cells that competes with the co-stimulatory molecule CD28 for its ligand expressed in antigen-presenting cells (APCs). Thus, CTLA-4 suppresses T cell response [14]. Since these CD28 ligands are expressed in APCs, CTLA-4-based T cell regulation occurs in peritumoral lymph nodes. The first ICI approved by FDA for cancer therapy was ipilimumab, an anti-CTLA-4 blocking antibody. Ipilimumab was evaluated in urothelial BC patients and showed an increase in CD4+ and CD8+ T cells in both tumor and blood, thus increasing inflammatory cytokine signature [15]. Another immune checkpoint molecule target for cancer therapy is programmed cell death protein (PD-1). PD-1 is expressed in T cells, and the interaction with its ligands PD-L1 and PD-L2 in normal cells inhibits T cell responses, restricting over-reactive T cells and hence autoimmunity [14]. In fact, the FDA has approved five PD-1/PD-L1 inhibitors in platinum-refractory metastatic urothelial carcinoma: atezolizumab, avelumab, durvalumab, nivolumab, and pembrolizumab. Pembrolizumab and atezolizumab have demonstrated clinical benefits with an objective response rate (ORR) ranging from 23% to 29% in patients ineligible for cisplatin [16,17]. Unfortunately, PD-1/PDL1 inhibitor treatment results in response in a minority of patients, showing decreased survival in patients with low PD-L1 expression. As a result, it is mandatory to identify those patients that are going to be sensitive to specific ICIs and also carry out alternative therapies alone or in combination with ICI. Recently, a phase Ib clinical study combining PD-1 blockade plus the personalized neoantigen-based vaccine NEO-PV-01 had been developed [18]. Cancer neoantigens are peptides unique to cancer cells that arise from tumor mutations and are important targets of T cell-mediated immunity [19]. The NEO-PV-01 vaccine induced neoantigen-specific CD4+ and CD8+ T cell responses in all patients. Interestingly, vaccine-induced T cells displaying cytotoxic phenotype were able to travel to the tumor and mediated cell killing [18].

Other non-cell-based immunotherapy strategies are being developed with promising results. For example, T cell-engaging bispecific antibodies (BiAbs). BiAbs bind to the tumor cell via tumor-associated antigen (TAA) and also to the T cell receptor CD3 subunit, inducing T cell recruitment and target cell killing [20]. As BiAbs engage endogenous T cells, it is not necessary to manipulate autologous T cells ex vivo to reinfuse them into patients, and therefore, it is a faster, cheaper, and non-patient-specific therapy. The major inconvenience of BiAbs in solid tumor treatment is their induction of tumor escape mechanisms such as TAA downregulation [21]. A potential option to avoid antigen escape is to combine bispecific antibodies to generate T cells that could recognize multiple antigens. Last year, a BiAb was designed that binds to B7-H3 tumor antigen and CD3 to treat T24 BC cells and xenograft mouse model in combination with trametinib, a MEK inhibitor [22]. Treatment with B7-H3-CD3 BiAb specifically and efficiently redirected T cell cytotoxicity against B7-H3 overexpressing tumor cells both in vitro and in vivo. Moreover, the combination of BiAb and MEK inhibitor increased T cell infiltration and significantly suppressed tumor growth. Another T cell-engaging strategy used for cancer immunotherapy is retargeting T cells to tumor cells by bispecific T cell engagers (BiTEs). BiTEs are small antibody-based proteins constructed of two single-chain variable fragments (scFvs) in tandem that physically link T cell to tumor cell [23]. However, as a non-full antibody molecule, BiTEs have a short serum protein half-life that forces the use of constant infusion pumps in the clinic. AMG 160 is a half-life extended BiTE that is having its safety and tolerability evaluated in metastatic castration-resistant prostate cancer patients in a phase I study (NCT03792841) [24].

## 3. Cell-Based Immunotherapies

In recent years, the optimization of technologies allowing efficient immune cell enrichment and expansion in vitro has been essential to deliver these cellular products into patients and apply cell-based immunotherapies in the clinic.

### 3.1. Dendritic Cells

Cell-based immunotherapies can target innate or adaptive immune cells. Naturally, immature dendritic cells (DCs) are able to take up exogenous antigens, migrate to lymph nodes, mature and present those antigens to T cells together with co-stimulatory signals, which triggers T cell activation. The most common clinical treatment using DCs in urological malignancies is ex vivo antigenic peptide loading followed by autologous infusion. In 2001, autologous DCs pulsed with tumor antigen melanoma-associated antigen 3 (MAGE-3), commonly expressed in advanced BC, was synthesized to bind specifically to HLA-A24. These loaded DCs generated tumor-specific cytotoxic T lymphocyte (CTLs) response against a MAGE-3-expressing bladder cancer cell line [25]. An optimal antigen loading method determined the potency of adaptive cell response and the outcome of the interaction. DCs, taking HY antigens directly from HY peptides, RNA, or cell lysates induced a poor immune response when compared with DCs incubated with irradiated-apoptotic HY expressing tumor cells that resulted in complete protection [26]. Signals provided to DCs by apoptotic cells substantially augment the potency of DC therapy. Another approach to enhance the cytotoxicity of T cells induced by DCs was the transfection of DCs with human secondary lymphoid-tissue chemokine (*SLC*) and human interleukin-2 (*IL-2*) genes [27]. Autocrine production of SLC and IL-2 by DCs promoted DC proliferation and cytotoxicity against BC cells that was induced by the co-culture of transfected DCs and T cells. Instead of DC modification, monocyte-derived DCs from NMIBC patients could be induced to mature DCs using a specific cytokine cocktail (IL-1β, tumor necrosis factor (TNF)-α, interferon (IFN)-α, IFN-γ, and polyinosinic:polycytidylic acid) resulting in α-type 1-polarized DCs (αDC1s) [28]. Autologous αDC1 loaded with the ultraviolet B (UVB)-irradiated human BC cell line T24 increased bladder cancer-specific CTL responses. In another study, apoptotic T24 cells (after treatment with cisplatin) were used to activate immature DCs in vitro [29]. Importantly, these activated DCs, when reinjected into mice, induced a cytotoxic effect that suppressed tumor growth even in mice with T24 cisplatin-resistant cancer cells-derived tumors. This DC therapy could be a great strategy for managing chemoresistance in BC patients. In a phase II trial involving human epidermal growth factor receptor 2 (HER2)+ urothelial cancer patients, associated with poor clinical outcomes [30], peripheral blood monocytes were pulsed with granulocyte-macrophage colony-stimulating factor (GM-CSF) linked to a recombinant HER2 peptide and then infused into patients. However, no statistical differences in OS were observed, and only patients with a low disease burden and no previous neoadjuvant chemotherapy had more favorable hazard ratios. Despite the non-positive results in this clinical study, it is possible to envisage that, in the future, more individualized strategies based on the identification of suitable neoantigens for DC vaccination of patients will be developed and used in BC management.

### 3.2. Natural Killer Cells

Natural killer (NK) cells are innate immune cells able to directly recognize and kill tumor cells. When the equilibrium between positive and negative signals is disrupted in a tumor cell as a consequence of NK activating ligand upregulation and loss of inhibitory signals, NK cells induce tumor cell lysis by granzymes and perforins or via apoptosis induction [12]. Autologous and allogenic NK adoptive cellular therapies are being investigated in patients with solid tumors and have demonstrated potential efficacy but also a few setbacks [31]. A major challenge is the difficulty involved in expanding autologous NK cells from cancer patients to obtain enough functional NK cells to be reinfused. Several possibilities have been explored, such as culture with cytokines or feeder cells, and differentiation from hematopoietic stem cells. Another major inconvenience of autologous NK cells is their poor response against tumor cells due to their low expression of NKG2D receptors even when they persist some weeks circulating in patients [32]. Although allogenic NK cell therapy had better clinical efficacy than autologous NK cell therapy in breast cancer in terms of tumor response, the number of circulating tumor cells, and immune function [33], several studies warn about side effects caused by autologous NK cells when they are infused at high doses and repetitive treatments. To overcome this issue, allogenic NK cells should be modified to improve their effect on tumor cells. Targeting high-affinity natural killer (t-haNK) derived from NK-92 cell line was engineered to express high-affinity CD16, endoplasmic reticulum-retained IL-2, and a PD-L1-specific CAR [34]. Irradiated PD-L1 t-haNK cells lysed 20 human cancer cell lines, including urogenital cancer cells, and inhibited the growth of engrafted bladder tumors in NOD-Scid IL2Rgamma null (NSG) mice. These promising results should encourage further clinical development of allogenic NK immunotherapy in combination with ICI treatments in BC patients.

### 3.3. T Cells

As previously stated, BCG immunotherapy is the most common treatment in NMIBC patients, and the immune response associated with its response has been well studied. Although BCG produces an anti-tumor environment affecting the innate immune system, it has been reported that BCG instillations in NMIBC patients also induce immune anti-tumor responses mediated by CD4+ T cells and CD8+ cytotoxic T lymphocytes. This suggests a key role of T cells in BC anti-tumor defense, leading to the exploration of possible ICI and BCG combinations or the use of ICI after BCG failure (clinical trials on Table 1) [35,36]. In fact, high CD3+ stroma T cell infiltration was associated with improved survival in stage pT1 BC [37], whereas intratumoral CXCR5+CD8+ T cells indicated an excellent prognosis in MIBC basal and stromal-rich subtypes [38]. Since an abundance of infiltrated CD8+ T cells is associated with improved survival in MIBC patients [39], the presence of tumor-associated macrophages (TAMs) and immunosuppressive regulatory T cells (Tregs) correlated with a poor prognosis in those MIBC patients, pointing out the influence of tumor microenvironment (TME) in therapeutic responses [40]. Due to the limited therapeutic options for advanced BC patients and the role of tumor infiltrated lymphocytes (TILs) in solid tumor overcome, some approaches based on adoptive cell therapy using TILs have been developed. Moreover, TILs from primary bladder tumors could recognize not only tumor-associated antigens but also neoantigens [41]. The first step in autologous TIL therapy was performed by TIL expansion using IL-2 from primary bladder tumors and then functionally selected by co-culture with autologous tumor and INF-γ measurement [42]. Furthermore, intravesical therapy with tumor-reactive T cells decreased bladder tumor growth in mice and increased T cell infiltration without lymphodepleting chemotherapy in orthotopic tumors [43]. However, ex vivo TIL expansion and reinfusion into patients required large surgical samples with enough TILs and appropriate technological facilities; therefore, T cell engineering for immunotherapy is being developed.

T cell genetic modification to redirect effector cells against specific tumor antigens could be done by T cell receptors (TCRs) or CARs [44]. T cell modification with TCR or CAR receptors requires several weeks (Figure 1). T cells have to be isolated from blood patient leukocyte apheresis samples using immuno-selective beads and then stimulated in a proliferative environment with IL-2 and/or anti-CD3 antibodies [45]. Activated T cells are transfected with the TCR/CAR construct using preferentially viral methods such as lentivirus and retrovirus that allow the integration and construction of persistent T cell population. Then, T cell clones are expanded in vitro until their re-infusion back into the patient, who must be lymphodepleted by chemotherapy treatment [46].

TCRs are natural receptors for antigen recognition presented via major histocompatibility complex (MHC) molecules on APCs. TCRs known to be reactive against a specific tumor antigen are usually obtained from TILs and cloned into T cells [47]. TCRs could only recognize peptides in an MHC context which implies several problems. First, the antigen is recognized with low binding affinity and could generate cross-reactivity and off-target toxicity. Second, MHC restriction excludes half of the patients that are not human leukocyte antigen (HLA)-A2 positive, in which antigen recognition context for TCR is typically developed [12]. Third, MHC dependency avoids T cell recognition via TCR in tumor cells with low or non-MHC expression, these tumors being resistant to TCR therapy.

CARs are designed to recognize a specific tumor antigen by their extracellular domain composed of a monoclonal antibody-derived scFvs [48]. A transmembrane domain is joined to the hinge of the extracellular domain to an intracellular signaling molecule comprised of the TCR CD3ζ signaling chain in first-generation CARs (Figure 1). Second and third-generation CARs incorporate co-stimulatory endodomains such as 4-1BB and CD28 alone or in combination, respectively, that allow them to survive and proliferate, which improves their engraftment and expansion within patients [12]. Recently, fourth-generation CAR-T cells were developed to enhance their efficacy by incorporation of cytokine overexpression, gene knock-out, and knock-in, targeting of multiple antigens simultaneously, and precise control of CAR expression and signaling [49]. A clear advantage of CARs as compared with TCR receptors is their high affinity due to their antibody character. Moreover, CARs can recognize several types of antigens—not only peptides but also glycosylation variants and non-peptide antigens expressed in the surface of tumor cells independently of the presentation via MHC by an APC cell. CAR binding to the tumor cell drives the activation of CAR-T cells and its direct effect against tumor cells.

Recently, CAR technology has been developed and applied in patients with hematological cancers with high rates of total remission [48]. However, CAR-T cell therapy in solid cancers had shown some difficulties, and the same successful results have not yet been obtained. To target tumor cells, firstly, CAR-T cells should migrate to tumor tissue which is usually highly impermeable because of the stromal architecture surrounding the tumor. Moreover, immunosuppressive TME affects CAR-T cells, reducing their effect against tumor cells. CAR-T cells should maintain and survive in hostile conditions, and it could require the use of high doses of CAR-T cells, as was demonstrated by Priceman et al. in a prostate orthotopic mouse model. In this study, prostate stem cell antigen (PSCA)-specific CAR T cells showed robust therapeutic efficacy in a subcutaneous prostate cancer model as compared with xenografts, highlighting the differences in solid TME and their impact on CAR-T efficacy [50]. However, an increase in CAR-T cell doses to reach efficacy could induce toxicity due to off-target effects since an ideal antigen exclusively expressed in cancer cells has not been identified yet for solid tumors [44]. One alternative has recently been reported that uses a chimeric PD1 (chPD1) receptor that recognizes the ligands for the PD-1 receptor that are expressed in many types of solid cancer [51]. The engagement of PD1 receptor to PD1 ligand-expressing tumor cells triggered T cell induction via CD3ζ activating domain and co-stimulatory receptor DAP10 that enhances T cell effector responses. chPD-1 T cells secreted pro-inflammatory cytokines and caused lyses in a MB49 BC cell line. Moreover, in a syngenic mouse model of BC, the tumor burden was significantly decreased in mice treated with chPD-1 T cells, among other types of solid cancer, and induced protective host anti-tumor memory responses.

CAR-T cell therapy has been tested in few clinical trials to treat urological malignancies [44]. Metastatic renal cell carcinoma was treated in 12 patients with autologous CAR-T cells against the carboxy-anhydrase-IX (CAIX) antigen that was expressed in cancer cells but also in bile duct epithelium cells, resulting in high toxicities and cessation of CAR-T treatment [52]. Subsequently, four patients were pre-treated with CAIX monoclonal antibody that blocked off-target antigenic sites in off-tumor organs, leading to the absence of liver toxicities. However, no clinical responses were recorded, although a first-generation CAR was used. In a later phase I clinical trial with five prostate cancer patients, CAR-T cells specific for prostate-specific membrane antigen (PSMA) were infused with no toxicities noted post-treatment [53]. Partial clinical responses were achieved in two patients, showing prostate-specific antigen (PSA) declined around 60% and PSA did not rise again for 150 days. Unexpectedly, clinical responses correlated inversely with T cell engraftment and directly with plasma IL-2 levels, suggesting that depletion of plasma IL-2 by a high number of activating CAR-T cells may limit clinical efficacy. Unfortunately, the new Pilot/Phase II trial (NCT01929239) planned to test moderate dose IL-2 together with high CAR-T engraftments for improved therapeutic efficacy was suspended due to lack of funding [54]. In 2019, Rosenberg and colleagues from National Cancer Institute concluded a clinical trial (NCT01218867) using CAR-T cells targeting anti-vascular endothelial growth factor receptor (VEGFR2) for patients with renal cancer with no objective responses, although promising results were obtained in several different models where VEGFR2 CAR-T cells inhibited tumor growth.

Several early phase clinical trials for T cell therapy of BC are currently active (clinicaltrials.gov accessed on 10 March 2021). A phase I/II study of the treatment of metastatic cancer that expresses MAGE-A3, including BC, using lymphodepleting conditioning followed by the infusion of anti-MAGE-A3 HLA-A*01 restricted TCR-gene engineered lymphocytes, and aldesleukin was not concluded due to insufficient accrual; therefore, no statistical results could be concluded (NCT02153905). A phase I/II study using fourth-generation CAR-T (4SCAR-T) cell therapies in advanced or metastatic urothelial BC patients who had no further treatment available is now in recruiting stage (NCT03185468). Fourth-generation CAR-T cells anti-PSMA or anti-Fos-related antigen (FRA) were evaluated in terms of side effects and effective doses in treating refractory and recurrent solid tumors. Another phase I study in recruiting stage was based on the combination of HER2-specific autologous CAR-T cell treatment with the injection of CAdVEC oncolytic adenovirus that was designed to help immune tumor response (NCT03740256). Finally, an active clinical study to evaluate CCT301-59 CAR-T cell therapy in patients with recurrent or refractory solid tumors, including BC, on the basis of safety, tolerability, and anti-tumor activity was started (NCT03960060). At the moment, there are several clinical studies in progress to test the efficacy and security of T cell therapy, alone or in combination with other therapies, in several solid tumors such as BC (Table 2).

#### 3.3.1. CAR-T Improvement

CAR-T cell therapies for severe and metastatic genitourinary cancer, including BC, with no other alternative treatments, could be a possible solution, although it is necessary to overcome some obstacles facing T cell therapy to get better and safer results [55]. Several strategies could be implemented to avoid off-target effects. For example, the selection of a scFv with decreased antigen affinity preserves anti-tumor effects while preventing its binding to normal tissues that express the antigen at low levels [56]. The most prevalent severe adverse effect after CAR-T infusion is a cytokine-release syndrome (CRS). A common choice to control T cell therapy toxicities is the use of inducible suicide genes in CAR construction to switch off CAR-T cells when treatment goes wrong. One CAR suicide gene is a truncated form of Caspase 9 that dimerized exclusively when a dimerizer molecule was added to trigger CAR-T cell apoptosis [45]. Inhibitory CARs (iCARs) are an alternative possibility to prevent off-target effects. Inhibition occurs when a second CAR receptor, with an inhibitory intracellular domain, recognizes a normal antigen in healthy tissue and negatively regulates the cytotoxic CAR effect [57]. Moreover, CARs that co-expressed two antigen receptors against two different TAA increased target specificity against tumor cells. Dual-antigen receptors based on Synthetic Notch (synNotch) receptor and CAR receptor are highly specific because they work in two steps. First, Notch receptors recognize tissue-specific antigens and releases a transcription factor that controls CAR expression. Next, the CAR receptor binds to tumor antigen, inducing CAR-T cell activation and cytotoxic effect.

Overcoming local immunosuppressive factors, such as inhibitory ligands and cytokines expressed by the tumor or regulatory host T cells, is necessary to improve CAR-T responses. So-called armored CAR-T cells, also known as fourth-generation CARs are CAR-T cells modified with a third co-stimulatory signal which improves their efficacy, expansion, and persistence as they are more resistant to an immunosuppressive TME [49]. CAR-T cells producing pro-inflammatory IL-12 prevented Treg cell inhibition via autocrine signaling by CAR-T IL-12 receptors [58]. However, overexpression of immune-stimulatory cytokines could induce severe side effects that should be addressed by the use of suicide genes or other safety strategies. The CAR-T cell response depends not only on strong activating signals but also on low inhibitory ones. For that reason, knock out of negative T cell regulators, like PD-1 disruption using CRISP/Cas-9 technology, enhanced anti-tumor activity of PSCA-CAR-T cells in NSG mice bearing established large PC3-PDL1 tumors [59]. PSMA-CAR-T cells modified with a dominant-negative tumor growth factor (TGF)-b type II receptor gene reduced the total burden of PSMA prostate cancer tumors in a mouse xenograft model due to their resistance to TGFb-mediated immune suppression [54]. It is widely known that T cell therapy combined with other anti-tumor treatments could be a good choice to obtain synergistic effects. Treatment with blockade antibodies against PD-1 or PD-L1 may strengthen CAR-T cell immune response as was described for HER-CAR-T cells enhanced activity in the presence of anti-PD1 antibody in pre-clinical models [60]. In the end, the best strategy for the enhancement of CAR-T cell function must be carefully studied in each cancer type, and even more, they may differ between different patients because of the idiosyncrasy of each tumor and anti-tumor immune response.

#### 3.3.2. CAR-T Drawbacks

CAR-T cell therapy usually induces CRS. After engaging with the corresponding target antigen, CAR-T cells produce inflammatory cytokines and chemokines, including IFN-γ, TNF-α, GM-CSF, IL-2, IL-8, and IL-10, which trigger the activation of host immune cells [61]. A potent immune response induces hyperinflammation and a positive CRS feedback loop. Common CRS symptoms are fever, hypotension, and hypoxia that could degenerate into respiratory failure, shock, and organ dysfunction if CRS is not treated in time. Another relevant acute toxicity caused by CAR-T cells is immune effector cell-associated neurotoxicity syndrome (ICANS) which is observed in 64% of clinical trials 4–5 days after CAR-T cell infusion [61]. The first signs of ICANS are confusion and aphasia, but it can progress to coma, seizures, and cerebral edema mostly associated with CRS. Both CRS and ICANS are generally reversible if treated promptly. Prevention of CRS and ICANS is a challenge in the treatment of patients with CAR-T cells. Several anti-cytokine and corticosteroid treatments together with CAR-T cell dosing and risk factors associated with these syndromes are being investigated. One of the main disadvantages of CAR-T cell therapy is its high costs. The estimated total cost of care associated with the administration of CAR-T cell therapy was $454,611 in an academic hospital inpatient setting [62]. Another study found that the median total cost of hospitalization resulting from CAR-T cell treatment was $380,052, with a median direct cost of $262,981 [63]. This cost of CAR-T cell therapy was calculated for a one-time treatment, but severe complications can easily push the total cost of care to $1 million. It is hoped that the next generation of CAR-T cell therapies will have fewer side effects and have a reduced price.

### 3.4. Other Cells

Cell types such as γδ T cells and Natural killer T (NKT) cells have recently drawn interest as innovative cellular cancer immunotherapies. Human γδ T cells have two main advantages for their anti-tumor use [12]. First, they are able to recognize and kill transformed cells independently of HLA restriction. Second, apart from T cell receptors, γδ T cells also express activating NK receptors, such as NKp30, NKp44, or NKG2D, which bind to stress-inducible surface molecules that are absent on healthy cells but are frequently expressed on malignant tumor cells [64]. Adoptive transfer of expanded γδ T cells seems to be safe in advanced prostate cancer and renal cancer, and partial remission or stable disease has been achieved.

NKT cells are a mixed population of NK and T cells that co-express an αβ T cell receptor in addition to cell surface markers of NK cells such as NK1.1, CD16, and CD56 [12]. NKT cells mediate their anti-tumor immune response by glycolipid recognition via CD1d presentation and subsequent enhancement of both innate and adaptive immune systems [65]. NKT cell activation triggers a prompt release of an array of cytokines, including IL-2, IFN-γ, TNF-α, and IL-4, which modulate different immune cells present in the TME, thus affecting anti-tumor immune responses. Although promising results have been obtained in some pre-clinical studies, the anti-tumor potential of NKT cells is being analyzed at the moment in several clinical trials that do not yet include BC.

Although CAR-NK and CAR-Macrophages (CAR-Ms) have not been used in the treatment of genitourinary cancer yet, both CAR-based cell therapies have shown remarkable results in other cancer types. The safety of CAR-NK cells was higher as compared with CAR-T cells, due in part to their limited lifespan in circulation and the fact that cytokines released by NK cells are not highly associated with CRS [66]. Moreover, allogenic CAR-NK cells have a reduced risk for graft versus host disease (GVHD). CAR-NK cells naturally have cytotoxic activity against tumor cells and can be activated through CAR-independent mechanisms, which adds to their CAR effect against tumor cells. In the case of CAR-Ms, their capacity to penetrate tumors, combined with their phagocytic activity, make them suitable cells for solid cancer therapy. In a study infusing CAR-Ms in solid tumor xenograft mouse models, CAR-Ms decreased tumor burden by antigen-specific phagocytosis and pro-inflammatory cytokine expression that polarize TME to an anti-tumor state [67].

## 4. Xenoinjection

Xenogeneic cell-based therapy consists of the implantation or infusion into human body fluids, tissues, or organs of viable somatic cell preparations of non-human animal cells as was defined by the European Medicines Agency (EMEA) in 2009. A common non-human animal cell source for xenogeneic cell therapy is pigs which are used to restore lost physiological tissue function and repair wounds caused by cancer [68]. Naturally, the product that is administered must be of acceptable quality and standards and free from contamination as was described in the guidelines for xenogeneic cell-based therapy medicinal products (European Medicines Agency (EMEA)/Comitee for Medicinal Products for Human Use (CHMP)/Cell-based Product Working Party (CPWP)/83508/2009).

The major drawback of xenotransplantation is the immunological rejection of the organ, tissue, or cell grafts [69]. This process is mediated in part by two antibodies called hyperacute rejection (HAR) and acute humoral xenograft rejection (AHXR) that attacks vascularized organs that have been transplanted from pigs. As a consequence, clinical xenotransplantation trials using xenogeneic cells that are not vascularized, instead of whole organs, prevented rejection [68]. Cellular xenotransplantation of urothelial cells into bladders in the clinic induced host immunological barriers and subsequent xenograft rejection. However, this immunological activation could be used as an advantage for anti-tumor immune response. In pre-clinical murine bladder tumor models, intravesical xenogeneic urothelial cell immunotherapy extended survival and repressed tumor progression by promoting T cell infiltration and activation [70]. Thus, xenogeneic urothelial cells triggering rejection induced T cell activation for anti-tumor activity.

## 5. Conclusions

As we have described in this review, several promising options for cell-based therapies of BC have been developed. Although most of the studies were performed using T cells and specifically CAR-T cells, other alternative treatments using innate immune cells such as DC or NK cells in pre-clinical models or even in patients were tested. CAR-T cell therapy, which is very effective in the treatment of blood cancers, showed different safety and efficacy drawbacks in solid tumors and in bladder cancer. The improvement of fourth-generation CAR-T cells and their use in combination with other treatments such as ICI, cytokines, or neoantigen-based vaccines will hopefully improve therapy response in BC patients in the future.

## Figures and Tables

**Figure 1 ijms-22-02818-f001:**
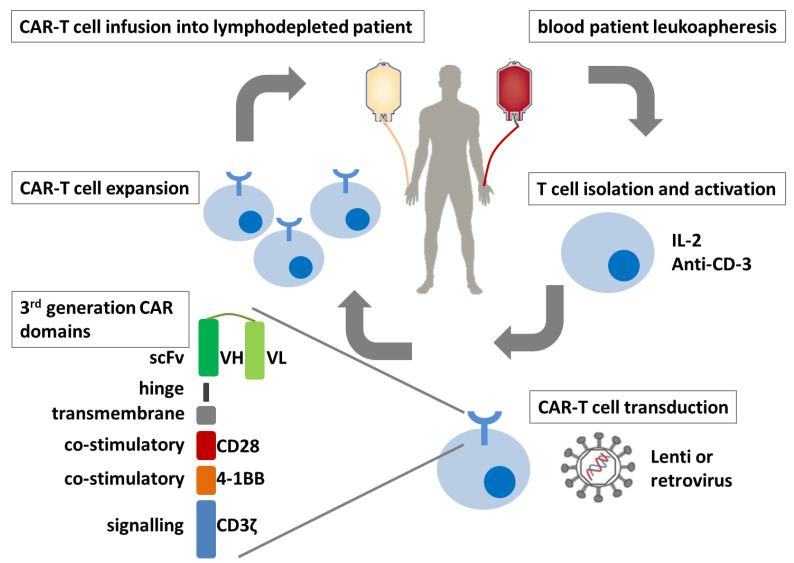
CAR-T cell therapy. Autologous T cells are modified with a third-generation CAR to be reinfused into the patient. CAR, chimeric antigen receptor; scFv, single-chain variable fragment VH, variable heavy chain; VL, variable light chain.

**Table 1 ijms-22-02818-t001:** Clinical trials evaluating ICIs in relation to BCG for NMIBC management.

Identifier ^1^	Title	Phase	Treatment
NCT03317158	ADAPT-BLADDER: Modern Immunotherapy in BCG-Relapsing Urothelial Carcinoma of the Bladder	I/II	BCG/ICI
NCT00539773	Phase II Trial of Concurrent Administration of Intravesical BCG and Interferon in the Treatment and Prevention of Recurrence of Superficial Transitional Carcinoma of the Urinary Bladder	II	BCG/IFN
NCT02901548	Phase 2 Durvalumab (Medi4736) for Bacillus Calmette-Guérin (BCG) Refractory Urothelial Carcinoma in Situ of the Bladder	II	ICI
NCT02808143	Pembrolizumab and BCG Solution in Treating Patients With Recurrent Non-Muscle-Invasive Bladder Cancer	I	BCG/ICI
NCT02844816	Atezolizumab in Treating Patients With Recurrent BCG-Unresponsive Non-muscle Invasive Bladder Cancer	II	ICI
NCT04164082	Testing the Addition of an Anti-cancer Drug, Pembrolizumab, to the Usual Intravesical Chemotherapy Treatment (Gemcitabine) for the Treatment of BCG-Unresponsive Non-muscle Invasive Bladder Cancer	II	Chem/ICI
NCT03106610	Trial of Anti-PD-1 (Nivolumab) in Bladder Cancer Patients Recently Treated With Intravesical BCG Immunotherapy	I	ICI
NCT03345134	Pembrolizumab in Combination With BCG After Ablation in Patients With UUTTCC Without Nephroureterectomy	II	BCG/ICI
NCT04134000	Atezolizumab and BCG in High-Risk BCG naïve Non-muscle Invasive Bladder Cancer (NMIBC) Patients (BladderGATE) (BladderGATE)	I	BCG/ICI

^1^ Identifier used by clinicaltrial.gov (accessed on 10 March 2021). BCG, Bacillus Calmette–Guerin. ICI, immune checkpoint inhibitor. IFN, interferon. Chem, chemotherapy.

**Table 2 ijms-22-02818-t002:** Ongoing T cell therapy clinical trials in bladder cancer.

Identifier ^1^	Phase	Engeneering T Cells	Tumor Antigen	Combined Treatment	Status
NCT02153905	I/II	TCR	MAGE-A3	Aldesleukin	Terminated
NCT03185468	I/II	CAR	PSMA and FRA		Recruiting
NCT03740256	I	CAR	HER2	CAdVEC ^2^	Recruiting
NCT03960060	I	CAR	ROR2		Active

^1^ Identifier used in clinicaltrial.gov (accessed on 10 March 2021). ^2^ CAdVEC oncolytic adenovirus; TCR, T cell receptor; CAR, chimeric antigen receptor; MAGE-A3, melanoma-associated antigen 3; PSMA, prostate-specific membrane antigen; FRA, Fos-related antigen; HER2, human epidermal growth factor receptor 2; ROR2, receptor tyrosine kinase like orphan receptor 2.

## Data Availability

Data sharing not applicable.

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
