# Peer review of "Cell Therapies in Bladder Cancer Management"

_ijms, 2021, doi:10.3390/ijms22062818_

Round 1

Reviewer 1 Report

In this manuscript the authors performed a review concerning the use of cell therapies against bladder cancer. This is an interesting manuscript with scientific sound but needs some improvments:

The authours should describe in introduction the treatments of invaive bladder cancer, namely the drugs used;

The authors should describe the side effects and costs of these therapies;

The xenoinjection most be better explained, with its advanatges and disadavantages

Author Response

Response to Reviewer 1.

1) The authours should describe in introduction the treatments of invasive bladder cancer, namely the drugs used.

We appreciate the reviewer’s suggestion. In the revised version we describe several current treatments of invasive BC in the introduction section. We also mention several treatments and drugs that can be used, such as partial cystectomy follow by chemoradiation, neoadjuvant dose dense MVAC treatment, immunotherapy using ICIs pembrolizumab or atezolizumab and other novel treatment options such as erdafitinib and enfortumab vedotin.

This information is included in the manuscript with colour as follows:

“For highly selected patients, a less aggressive partial cystectomy followed by chemoradiation is an alternative that may provide similar oncologic outcomes while maintaining bladder and sexual functions [4]. European association of urology (EAU) guidelines on muscle-invasive and metastatic BC consider adjuvant chemotherapy treatment after surgery if patients do not receive previously neoadjuvant [5]. Recently, neoadjuvant dose dense methotrexate, vinblastine, doxorubicin, and cisplatin (MVAC) treatment demonstrated improved survival rates in patients with locally advanced BC as compared with gemcitabine and cisplatin (NCT01031420) [6].” Page 1, lines 19-26.

“In platinum-relapsed patients with metastatic urothelial carcinoma, immunotherapy treatment using ICIs pembrolizumab or atezolizumab are the second-line treatment op-tion [9,10],” Page 2, lines 35-37.

“Novel treatment options for these patients were approved by Food and Drug Administration (FDA) and are currently under clinical investigation. Erdafitinib is a pan-fibroblast growth factor receptor inhibitor that targets this signalling pathway involved in BC tumorogenesis and enfortumab vedotin is an antibody-drug conjugate (ADC) therapy that recognizes bladder cancer cells to deliver cytotoxic drugs [11].” Page 2, lines 45-46.

2) The authors should describe the side effects and costs of these therapies.

We agree with the reviewer that this is a potential important aspect. To point out the cost of monitoring NMIBC patients after their treatment, we include the following sentence in the introduction section: “These NMIBC recurrence rates imply a thorough monitoring after their treatment for an extended period which is associated with a high cost for health care systems.” Page 1, lines 15-16.

We discuss side effects and cost of CAR-T therapy in a new section in page 9 called 3.3.2. CAR-T drawbacks. We get indeed in cytokine-release syndrome and immune effector cell-associated neurotoxicity syndrome as the most extended and serious side effects. We describe CAR-T therapy cost for health care systems in several studies.  

“3.3.2. CAR-T drawbacks

CAR-T cell therapy usually induce CRS. After engaging with the corresponding target antigen, CAR-T cells produce inflammatory cytokines and chemokines including IFN-γ, TNF-α, GM-CSF, IL-2, IL-8 and IL-10 which trigger the activation of host immune cells [61]. A potent immune response induces hyperinflammation and a positive CRS feedback loop. Common CRS symptoms are fever, hypotension and hypoxia that could degenerate into respiratory failure, shock and organ dysfunction if CRS is not treated in time. Another relevant acute toxicity caused by CAR-T cells is immune effector cell-associated neurotoxicity syndrome (ICANS) that is observed in 64% of clinical trials 4-5 days after CAR-T cell infusion [61]. The first signs of ICANS are confusion and aphasia but it can progress to coma, seizures and cerebral edema mostly associated with CRS. Both CRS and ICANS are generally reversible if treated promptly. Prevention of CRS and ICANS it is a challenge in the treatment of patients with CAR-T cells. Several anti-cytokine and corticosteroid treatments together with CAR-T cell dosing and risk factors associated with these syndromes are being investigated.

One of the main disadvantages of CAR-T cell therapy is its high price. The estimated total cost of care associated with the administration of CAR T-cell therapy was $454,611 in the academic hospital inpatient setting [62]. Another study found that the median total cost of hospitalization resulting from CAR-T cell treatment was $380,052, with a median direct cost of $262,981 [63]. This cost of CAR-T cell therapy is calculated for a 1-time treatment, but severe complications can easily push the total cost of care to $1 million. It is hoped that the next generation of CAR-T cell therapies will have fewer side effects and will reduce its price.”

3) The xenoinjection most be better explained, with its advanatges and disadavantages

To address this issue, the beginning of xenoinjection section in page 10 has been rewritten to better define what is xenogeneic cell-based therapy and which are their advantages as compared with other similar approaches such as xenotransplantation of organs.

  1. Xenoinjection

“Xenogeneic cell-based therapy consists of the implantation or infusion into human body fluids, tissues or organs of viable somatic cell preparations of non-human animal cells as was defined by the European Medicines Agency (EMEA) in 2009. A common non-human animal cell source for xenogeneic cell therapy is pigs which are used to re-store lost physiological tissue function and repair wounds caused by cancer [68]. Natu-rally, the product that is administered must be of acceptable quality and standard, and free from contamination as was described in guideline on xenogeneic cell-based therapy medicinal products (MEA/CHMP/CPWP/83508/2009).

The major drawback of xenotransplantation is the immunological rejection of the organ, tissue or cell grafts [69]. This process is mediated in part by two antibodies called hyperacute rejection (HAR) and acute humoral xenograft rejection (AHXR) that attack vascularized organs that have been transplanted from pigs. As a consequence, clinical xenotransplantation trials using xenogeneic cells that are not vascularized instead of whole organs prevented rejection [68]. Cellular xenotransplantation of urothelial cells into bladders in the clinic induced host immunological barriers and subsequent xenograft rejection. However, this immunological activation could be used as an advantage for an-ti-tumor immune response. In preclinical murine bladder tumor models, intravesical xenogeneic urothelial cell immunotherapy extended survival and repressed tumor pro-gression by promoting T cell infiltration and activation [70]. Thus, xenogeneic urothelial cells triggering rejection induced T cell activation for anti-tumor activity.”

Reviewer 2 Report

Morales and Paramio present a comprehensive review of cell based immunotherapies for bladder cancer. The review is well structured but the english writing need to be checked. Other than that, there are no issue that need to be addressed and the review can be published after a thorough language check.

As an optional point, this reviewer would have preferred slightly more focus on potential biomarkers (in development), both in the section for BCG and ICI, and potentially also (if they exist) for cell-based therapies against solid tumors, but it is understood that adding much information on response biomarkers for each section would take up too much space. If the authors deem it to be adequate, they may consider adding a brief state of the art regarding predictive biomarkers for each modality covered.    

Author Response

Response to Reviewer 2.

1) English writing need to be checked.

As Reviewer 2 asked, an English reviewer who is specialized in scientific issues has checked the manuscript. Throughout the manuscript we correct grammar, spelling and prepositions. Moreover, some sentences have been rewritten to be better understood. These corrections are coloured in the manuscript.

2) Optional point predictive biomarkers for BC therapies.

We agree this could be a very relevant aspect. However, at the present there are no reliable and confirmed reports regarding biomarkers in cell-based therapies against solid tumors, probably due to the low number of studies addressing these aspects. Accordingly, we decided do not include such information in the revised manuscript.

Round 2

Reviewer 1 Report

The authors performed the requested corrections and improved the manuscript scientific quality.